# The role of urbanisation in affecting *Mytilus galloprovincialis*

**Puri Veiga** [1,2] *, **Catarina Ramos-Oliveira** [1,2], **Leandro Sampaio** [1,2], **Marcos Rubal** [1,2]

**1** CIIMAR Interdisciplinary Centre of Marine and Environmental Research of the University of Porto, Novo Edifício do Terminal de Cruzeiros do Porto de Leixões, Avenida General Norton de Matos, Matosinhos, Portugal, **2** Department of Biology, Faculty of Sciences, University of Porto, Porto, Portugal

* puri.sanchez@fc.up.pt

**Data Availability Statement:** All relevant data are within the manuscript and its Supporting Information files.

**Funding:** This research was developed under the Project No. 30181, co-financed by COMPETE 2020,

## Abstract

Urbanisation is considered as one of the most critical and widespread threats to coastal marine ecosystems. The aim of this study was to compare the density, percentage cover, thickness of clumps, condition index and size-frequency distribution of *Mytilus galloprovincialis* between urban and non-urban shores, at nested spatial scales, in the Northern Portuguese coast. *M. galloprovincialis* was selected as model because it is economically and ecologically relevant. Moreover, the relationship between mussel size and the other variables (i.e. density, percentage cover, thickness and condition index) were investigated. Mussels on urban shores showed a smaller density and a greater frequency of larger individuals. A significant negative correlation between mussel length with density and with thickness of clumps was also found. Our results seem to indicate that recruitment has declined on urban shores and, as a result, intraspecific competition is also smaller, leading to more resources being available for a fewer individuals which can reach larger sizes. As mussel beds support a great biodiversity of invertebrates and provide many ecosystem services, urbanisation may have indirect effects on communities associated with mussels. Understanding the vulnerability of mussel beds to urbanisation could inform management.

## Introduction

Marine ecosystems provide food provision, natural shoreline protection against storms and floods, water quality maintenance, support of tourism and other cultural benefits, and maintenance of our societies [1,2]. Despite of their importance, marine ecosystems have been strongly modified, degraded or lost as consequence of anthropogenic activities [1]. Among anthropogenic disturbances, coastal urbanisation is one of the most pervasive and growing threats [3]. Human population density within 100 km of the coast is nearly three times greater than the global average [4]. Coastal areas favour the concentration of human populations because marine environment facilitates activities such as fishing, industry, tourism and transportation among other reasons [3]. Urbanisation of coastal areas is associated to different impacts as consequence of three primary interacting drivers: exploitation of living and non-living resources, pollution pathways, both industrial and domestic, and the proliferation of coastal and offshore artificial structures such as seawalls, jetties, piers or breakwaters [4]. In this way, urbanisation is a multifaceted, heterogeneous and complex phenomenon and it is highly

Portugal 2020 and the European Union through the ERDF, and by FCT through national funds. The funders had no role in study design, data collection and analysis, decision to publish, or preparation of the manuscript.

**Competing interests:** The authors have declared that no competing interests exist.

contextual [5]. Its ecological impacts are extreme and often irreparable such as habitat loss, spread of invasive species, disappearance of foundation species, changes in biodiversity, productivity and community composition, settlement of ruderal species and proliferation of jellyfish and toxic algae [4,6–9]. Urban impacts have been deeply evaluated on terrestrial and freshwater ecosystems, showing big modifications on their structure and function [4,10]. In the marine realm, most of the studies assessing the impact of urbanisation are focused on the effect of artificial structures on biodiversity [e.g. 9,11,12] and their role on the spread and settlement of non-indigenous species by comparing natural *versus* artificial habitats [e.g. 7,13]. Predictions indicate that human population living on the coast will double over the next decades, expecting that in 2025 nearly 75% of the global population will inhabit coastal areas [14]. However, understanding the effects of urbanisation on the structure and functioning of marine ecosystems has still been disregarded in the framework of conservation and management issues [15].

The Mediterranean mussel *Mytilus galloprovincialis* Lamarck 1819 is a widespread filter-feeding bivalve along the Atlantic rocky shores of the Iberian Peninsula, with a relevant role in intertidal food webs [16]. It is considered an ecosystem engineer species because it maintains useful habitat for other organisms, enhancing the biodiversity [17,18]. This species has long been harvested for food and bait and is now severely exploited in many European countries as Italy, Spain and Portugal [19]. Mussel harvesting has two adverse effects. Firstly, removal of adult mussels unavoidably eliminates discarded juvenile mussels. Secondly, mussel beds are the preferred settlement areas of their own recruits. Therefore, loss of mussels not only depletes the adult stocks, but also diminishes recruitment and slows down the recovery of mussels [20]. Mussels provide 13% of the global production of marine bivalves [21] but their farming depends on wild populations because young mussels are captured in natural systems and then deployed at culturing sites to grow-out [22]. Mussel beds are particularly vulnerable to anthropogenic disturbances [23]. For example, in 1990 all intertidal beds of *M. edulis* Linnaeus 1758 in the Dutch Wadden Sea were removed because of three consecutive years of recruitment failure, intensive fishery and great rates of natural mortality [24]. A significant negative exponential relationship was also found between density of *M. galloprovincialis* recruits and harvesting intensity, with intensities greater than 30% dramatically reducing the mussel recruitment and this pattern remained constant over 2 years [25]. Trampling and removal of *M. californianus* for bait by fisherman also diminished cover, density, biomass and size of mussels in California [26]. *M. galloprovincialis* is a species ecologically and economically relevant providing many ecosystem services as supporting (habitat for species, lifecycle maintenance, biodiversity), provisioning (food), regulating (water filtration, coastal protection) and cultural services (recreational fishing, symbolic) [27–29]. As mussel beds contribute to ecosystem function and the delivery of ecosystem services, understanding of their sensitivity to urbanisation may help to its management.

Considering that disturbances such as harvesting or trampling are usually more intense in urban coastal areas [26,30,31], the aim of this study was to compare density, percentage cover, thickness of clumps, condition index and size-frequency distribution of *M. galloprovincialis* between urban and non-urban shores in the Northern Portuguese coast. Moreover, the relationship between mussel size and the other studied attributes (i.e. density, percentage cover, thickness and condition index) was explored to disentangle potential effects of intraspecific competition.

## Material and methods

### Study area

The study was carried out in January 2019 at six rocky shores with different degree of urbanisation in the North West coast of Portugal. Field researches in this area were done in the frame

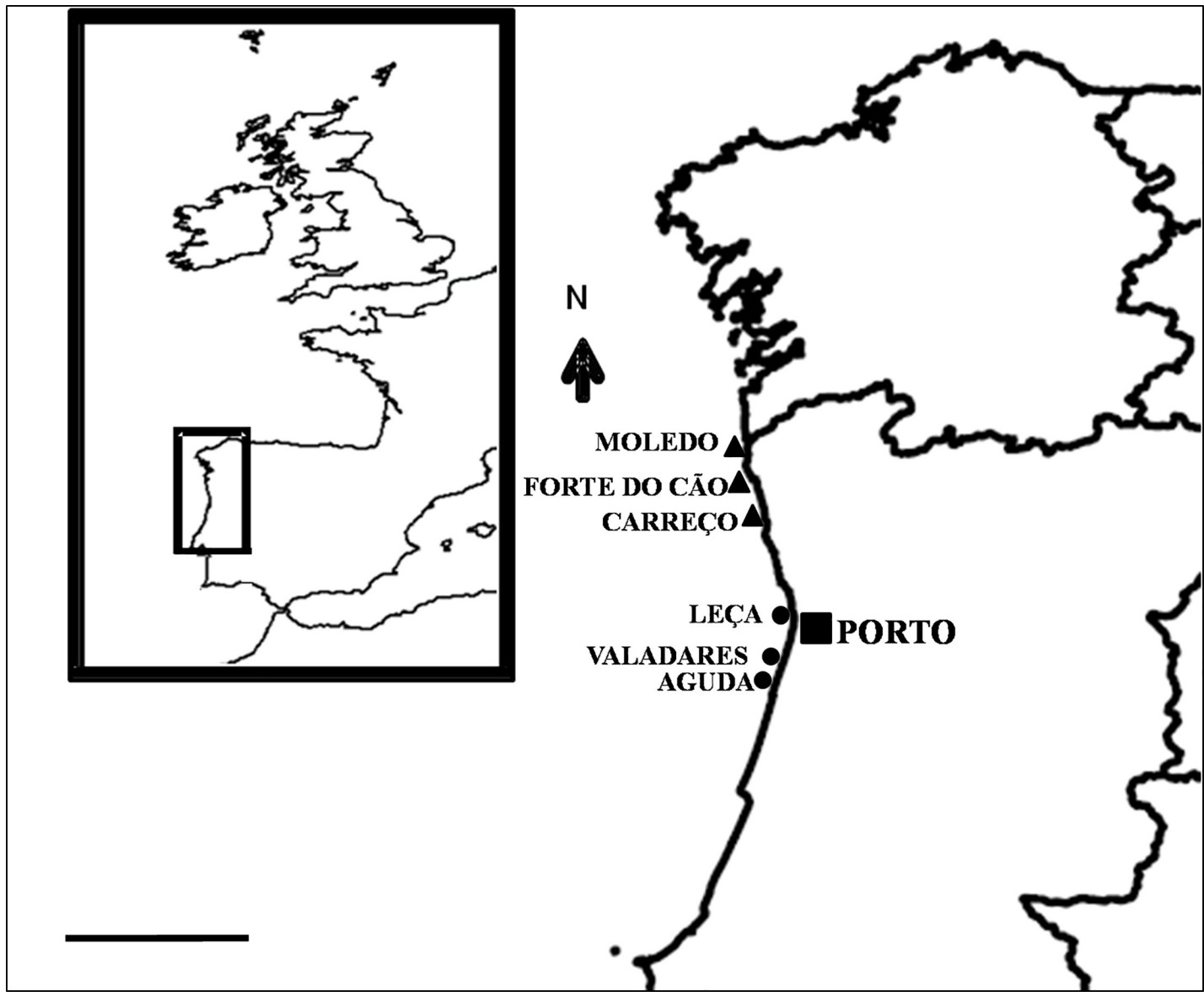

**Fig 1. Location of the rocky shores.** Scale bar 50 km.

of the project PTDC/CTA-AMB/30181/2017. No specific permissions were required for these locations or activities because the species is not protected or regulated for research activities. The field studies did not involve endangered or protected species or areas. Population density was used as a proxy of urbanisation because a greater population density relates to increased threats to marine ecosystems [32]. In the study area, population is mainly concentrated near the metropolitan area of Porto, the second largest Portuguese city (S1 Fig). Three rocky shores were considered as urban: Cabo do Mundo (41.225741 N 8.717976 W), Valadares (41.089167 N 8.658374 W) and Aguda (41.04246 N 8.653254 W). Moledo (41.822605 N 8.874894 W), Forte do Cão (41.798244 N 8.88748480 W) and Carreço (41.742040 N 8.878418 W) were considered as non-urban shores since they are located in areas near small towns (Fig 1 and S1 Fig).

These rocky shores are characterised by granitic substrate. The tidal regime is semidiurnal, with the largest spring tides of 3.5–4.0 m. The study was done on the mid-shore, which is dominated by the mussel *Mytilus galloprovincialis* and the barnacles *Chthamalus stellatus* (Poli, 1791) and *C. montagui* Southward 1976. Seaweeds are frequently represented by species such as *Lithophyllum* spp., *Corallina* spp., *Gelidium* spp., *Chondracanthus teedei* (Mertens ex Roth) Kützing and *C. acicularis* (Roth) Fredericq, *Ulva* spp. and *Bifurcaria bifurcata* R. Ross [33] commonly found in tidal-pools [34].

## Sampling and sample processing

A mixed model design was followed to compare mussel attributes between urban and non-urban shores at nested spatial scales. Two conditions (urban and non-urban) were considered. At each condition, three different rocky shores were chosen separated by 10s of km (Fig 1). Urban and non-urban sampling shores could not be interspersed because most of the population, commercial and industrial activities are concentrated near the Porto district (Fig 1 and S1 Fig). Previous studies [33,35–37] have indicated clear differences in pollution profiles between the shores assigned to each condition in our study. Moreover, a previous study done in northern Portugal [38] found no significant variability on rocky shore assemblages among locations separated by a similar spatial scale as urban and non-urban locations (10s of km). Therefore, any possible confounding factor due to the spatial segregation of urban and non-urban shores was likely irrelevant compared to the specific effects under examination.

At each of the three urban and non-urban rocky shores, two sites were randomly selected (about 10 m apart). At each site, the percentage cover of *M. galloprovincialis* was estimated in four quadrates (50 x 50 cm). Percentage cover estimates were obtained by dividing each quadrate into 25 subquadrates of 6 x 6 cm, assigning a score from 0 (absence of mussel) to 4 (a whole sub-quadrate covered by mussel) and adding up the 25 estimates [39]. When the mussel covered fewer than one subquadrate, an arbitrary value of 0.5 was assigned. Additionally, four random measures of mussel clump thickness were estimated for each sampled quadrate. Moreover, four quadrats (10 x 10 cm) were sampled by scraping off all mussels and samples were stored in a labelled plastic bag and frozen until further processing. In the laboratory, the number of mussels at each replicate was counted to estimate mussel density. The condition index was determined as the ratio between soft tissue dry weight and the shell dry weight at 10 mussels per quadrat (10 x 10 cm). Moreover, the shell length was measured in 20 mussels per quadrat (±0.1 mm) with a calliper, and each individual was assigned to specific size classes of shell length (Class 1: <10 mm, Class 2: 10–20 mm, Class 3: 20–30 mm, Class 4: 30–40 mm, Class 5: 40–50 mm and Class 6: >50 mm).

## Data analyses

Analyses of variance (ANOVA) were done to test for differences in the density and percentage cover of mussels among urban and non-urban shores. These analyses were based on a three-way model with condition as a fixed factor with two levels (non-urban and urban), shore as a random factor nested in condition with three levels and site randomly nested both in condition and shore with two levels and four replicates. ANOVA was also used to test for differences in the thickness of clumps and condition index. These analyses were based on a four-way model, including the same factors described above for density and percentage cover plus quadrat as an additional random factor nested in condition, shore and site with four levels, and four and ten replicates for thickness and condition index, respectively.

Cochran's C tests were previously done to check for homogeneity of variances, and when test was significant (p < 0.05) data were transformed to remove heterogeneity. When this was

**Table 1. Summary of ANOVAs for density and percentage cover of *M. galloprovincialis*.**

| Source of variation | df | Density | | % Cover | |
|---|---|---|---|---|---|
| | | MS | F | MS | F |
| Condition | 1 | 3742508.52 | 7.32* | 660.08 | 0.55 |
| Shore | 4 | 511616.15 | 0.96 | 1196.79 | 5.12* |
| Site | 6 | 535583.85 | 2.05 | 236.67 | 3.04* |
| Residual | 36 | 261227.05 | | 76.94 | |
| Total | 47 | | | | |
| Transformation | | none | | none | |
| Cochran's test | | C = 0.32 | ns | C = 0.27 | ns |

ns: not significant

* p<0.05

not possible, untransformed data were analysed and results were considered robust if significant at p < 0.01, to compensate for the increased probability of type I error [40]. Whenever ANOVA showed significant differences (p < 0.05), a post hoc Student-Newman-Keuls (SNK) test was done to explore differences between conditions.

To analyse differences in mussel size between urban and non-urban shores, their size-frequency was compared by means of Kolmogorov–Smirnov tests (KS).

In order to explore the relationship between mussel size and the other mussel attributes (i.e. density, percentage cover, thickness of clumps and condition index), rank correlation analyses were done. Due to the non-normal distribution of the data, Spearman's rank correlation was used.

## Results

### Effects of urbanisation on mussel attributes

Density of mussels was significantly lower in urban shores with urban shores showing a value of density nearly half of that at non-urban shores (Table 1, Fig 2A). Significant differences between conditions (i.e. urban versus non-urban) were not detected for the percentage cover,

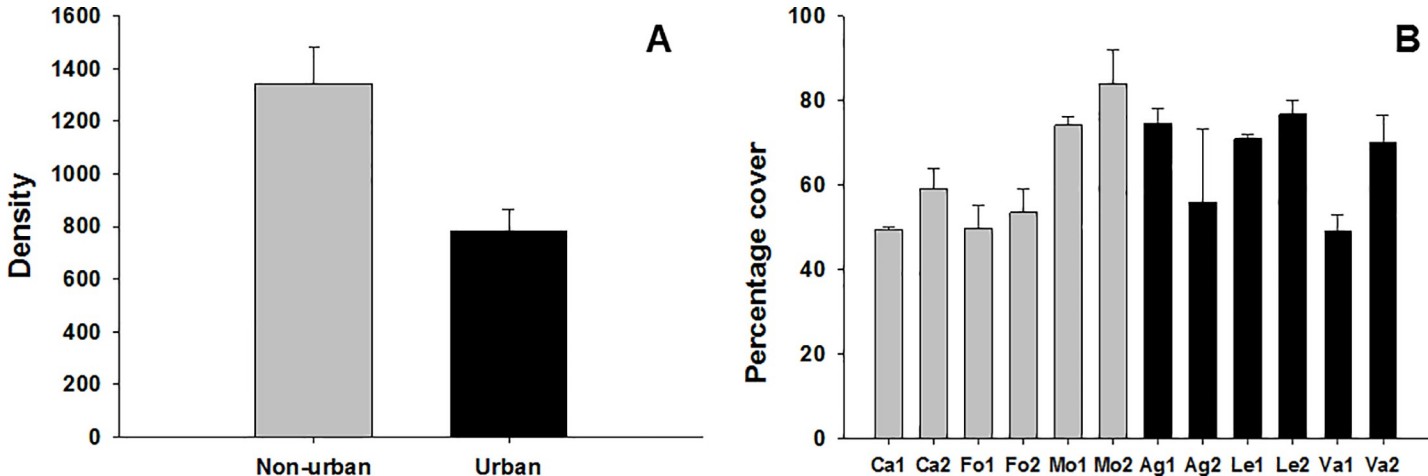

**Fig 2. Density and percentage cover of mussels at urban and non-urban shores.** Mean values (+ SE) of density (number of mussels per 10 cm²) (A) and percentage cover in 50 x 50 cm (B).

**Table 2. Summary of ANOVAs for clump thickness and condition index of mussels.**

| Source of variation | Thickness of clumps | | | Condition index | | |
|---|---|---|---|---|---|---|
| | df | MS | F | df | MS | F |
| Condition | 1 | 0.56 | 0.52 | 1 | 0.008 | 1.16 |
| Shore | 4 | 1.07 | 5.94* | 4 | 0.007 | 4.36 |
| Site | 6 | 0.18 | 1.74 | 6 | 0.0015 | 0.74 |
| Quadrat | 36 | 0.10 | 3.09*** | 36 | 0.002 | 2.10*** |
| Residual | 144 | 0.03 | | 432 | 0.001 | |
| Total | 191 | | | 479 | | |
| Transformation | | Sqrt(X+1) | | | none | |
| Cochran's test | | $C = 0.10$ | ns | | $C = 0.62$ | s |

ns: not significant; s: significant

*: p<0.05

*** p<0.001

thickness of clumps and condition index of mussels (Tables 1 and 2). However, these variables showed significant variability at the scale of shore and site (percentage cover), shore and quadrat (thickness of clumps) and quadrat (condition index) (Tables 1 and 2; Figs 2B and 3).

In terms of size, significant differences between urban and non-urban shores were detected (KS test, Dmax = 5.9, p < 0.001). The number of mussels included in greater size classes was significantly larger at urban (Fig 4A) than at non-urban shores (Fig 4B).

## Relationship between size and mussel attributes

Spearman's rank correlations showed that mussel density and thickness of clumps significantly decreased with mussel size, estimated as shell length (Fig 5A and 5C). However, there was no significant relationship between mussel size and percentage cover and condition index (Fig 5B and 5D).

## Discussion

Disturbances associated with coastal urbanisation are consistently considered as the most severe and prevalent threats to global marine ecosystems [41]. Many studies estimate that population density in coastal areas will increase in the future and consequently also the coastal urbanisation [14]. The effect of urbanisation on marine resources is hardly reflected in the frame of urbanisation [4] and to fill this gap our study compared different attributes of *M. galloprovincialis*, an exploited commercial species with a relevant ecological role, between urban and non-urban conditions. The most notorious result of our study was that mussels at urban shores showed a smaller density and a greater frequency of larger individuals. In freshwater habitats, different studies have also evaluated the effects of urbanisation on mussel species. In concordance with our results, Gillis et al. [42] found that urbanisation reduced the abundance of freshwater bivalves and increased the frequency of larger individuals of *Lasmigona costata*, the most abundant species in their study. Previously Gillis [43] found that the same *L. costata*, showed a smaller condition factor downstream of the urban area (more urbanised) than upstream of the cities (less urbanised). This contrasts with our results since we did not find significant differences between urban and non-urban shores for condition index. Nevertheless, for this attribute, we found significant variability at the scale of quadrat (between meters). The study by Gillis [43] did not consider nested spatial scales and this reflects the importance of an appropriate scale of spatial replication.

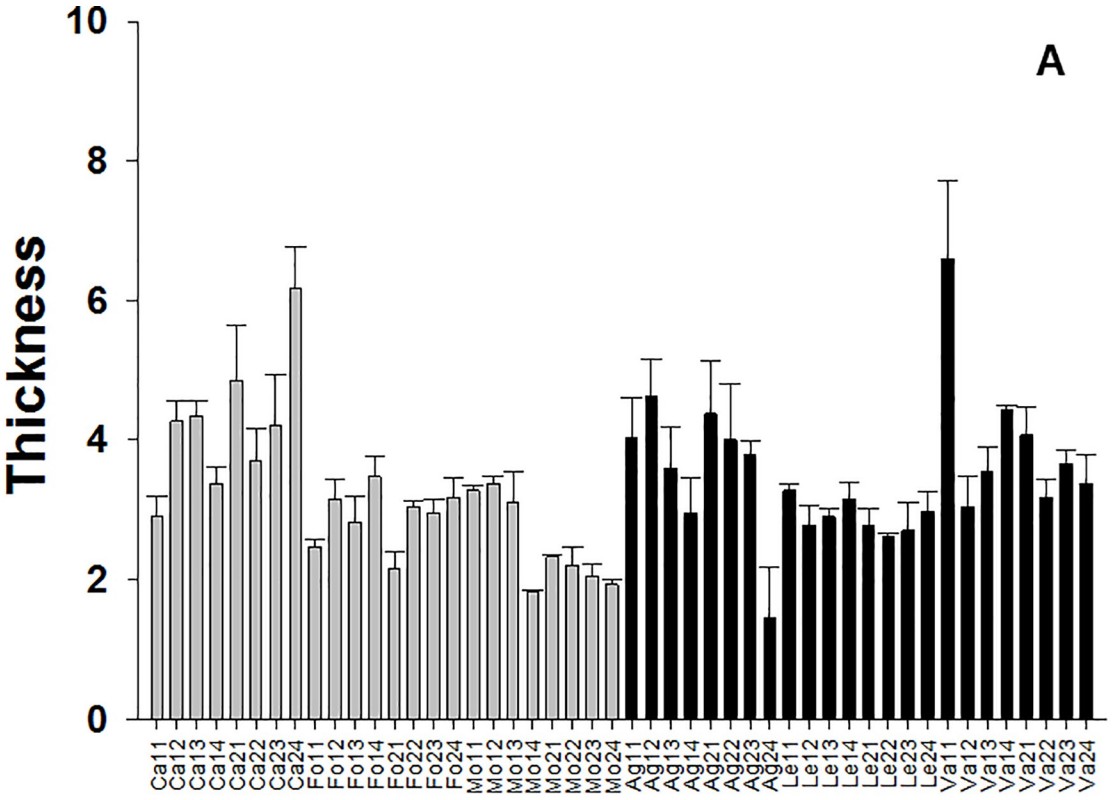

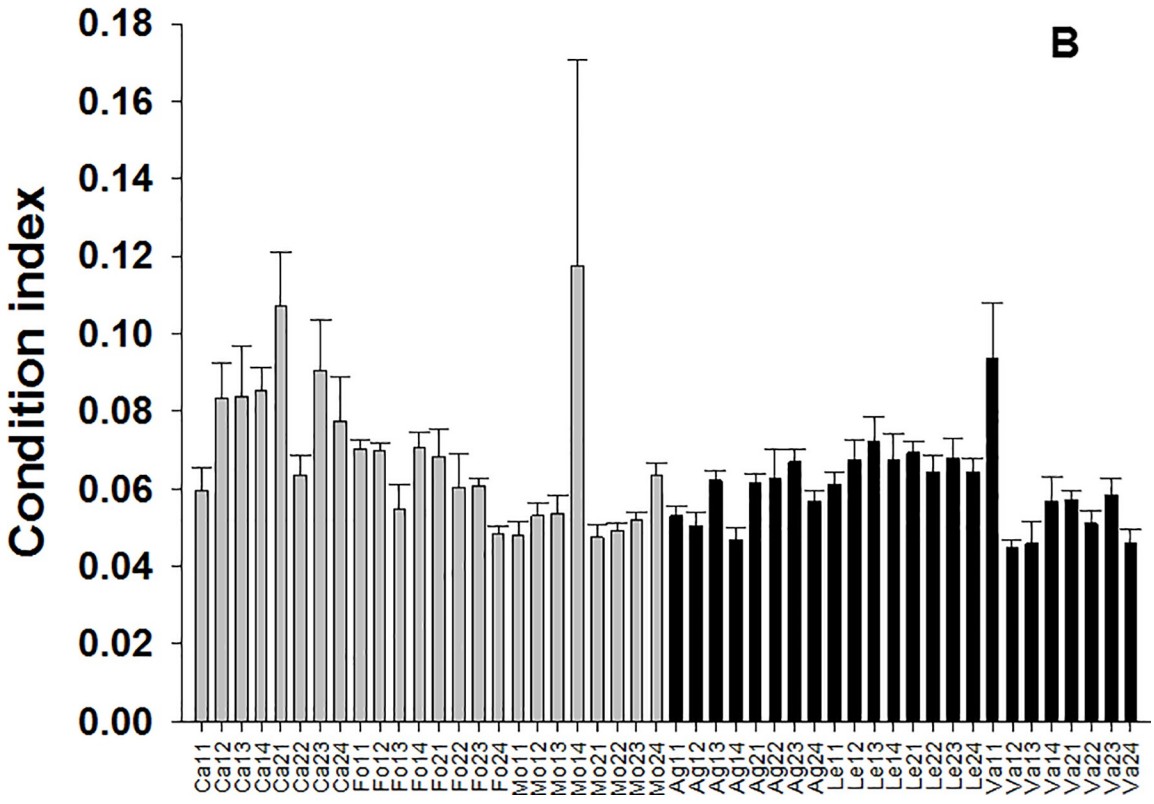

**Fig 3. Thickness of clumps and condition index of mussels at urban and non-urban shores.** Mean values (+SE) of thickness of clumps in cm (A) and condition index (B).

In the marine realm, different works have proved that mussel beds are particularly vulnerable to anthropogenic disturbances. Rius and Cabral [44] found a strong decline in density of *M. galloprovincialis* at more accessible sites to humans, frequently associated with more urban areas. Carranza et al. [23] found that population of *Mytella strigata*, an estuarine mussel species in South America, declined in urbanised areas. *Mytilus edulis* showed to be sensitive to introduction or spread of non-indigenous species, habitat structure changes such as removal of substratum (extraction) and physical loss (to land or freshwater habitat), impacts typically associated with urbanisation [45]. Smith et al. [46] assessed the influence of human presence on different attributes of *M. californianus* populations in California. They found that percentage cover, bed thickness and biomass was smaller at sites subjected to greater levels of human activities or presence. Their results [46] contrast with our observations because we did not find significant differences on these variables. This inconsistence may be due to a different responses of the two species of *Mytilus* or because California populations are submitted to a larger intensity of disturbance than our study area. Other previous studies found that intense disturbances reduced the cover of mussels, for example, Airoldi and Bulleri [47] studied the effects of urban infrastructures (breakwaters), concretely their maintenance that involves the addition of new quarried rocks over large portions of the defence structures to repair harms from storms. They found that maintenance interventions to breakwaters produced a significant decrease in the cover of *M. galloprovincialis* principally on the landward (sheltered) sides of breakwaters where maintenance was harsher and continual. Moreover, different studies found that the structure of dominant mussel beds also differs between the landward and seaward sides of breakwaters. On the landward sides, mussels are generally larger-sized and form a multi-layered matrix whilst on the seaward sides individuals are smaller-sized and form mono-layered beds, which tend to be less susceptible to dislodgement by physical disturbances [13]. We did not find significant differences on the thickness of mussel clumps between urban and non-urban shores. However, we found a significant negative correlation between mussel

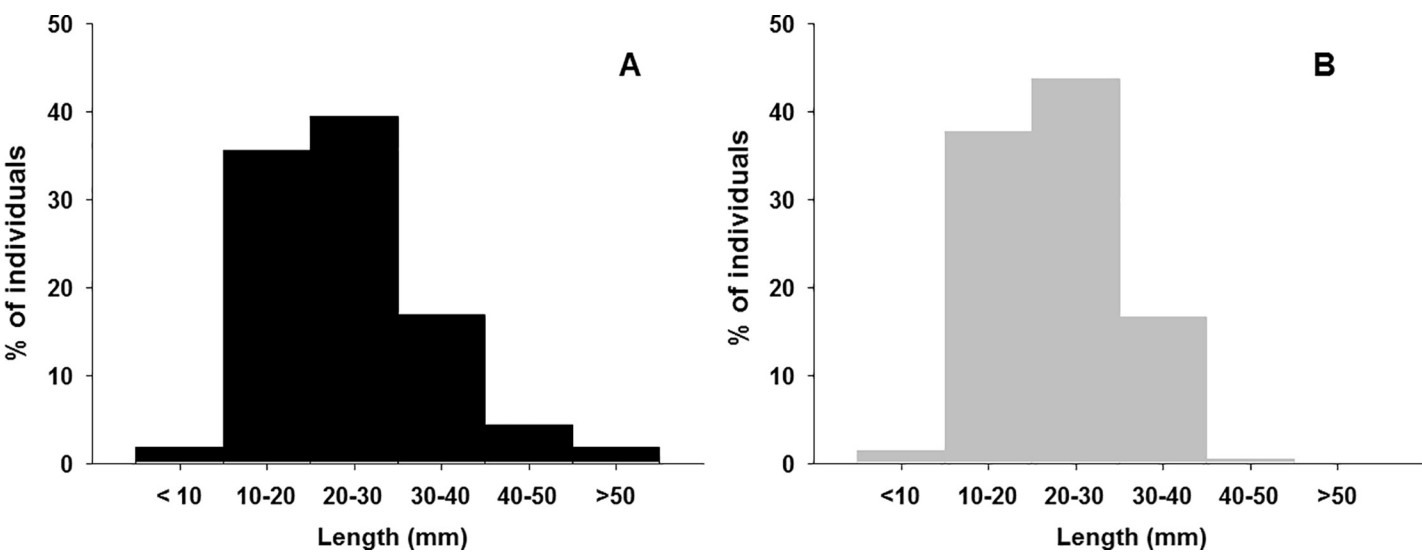

**Fig 4. Percentage of mussels per size classes.** At urban (A) and non-urban shores (B).

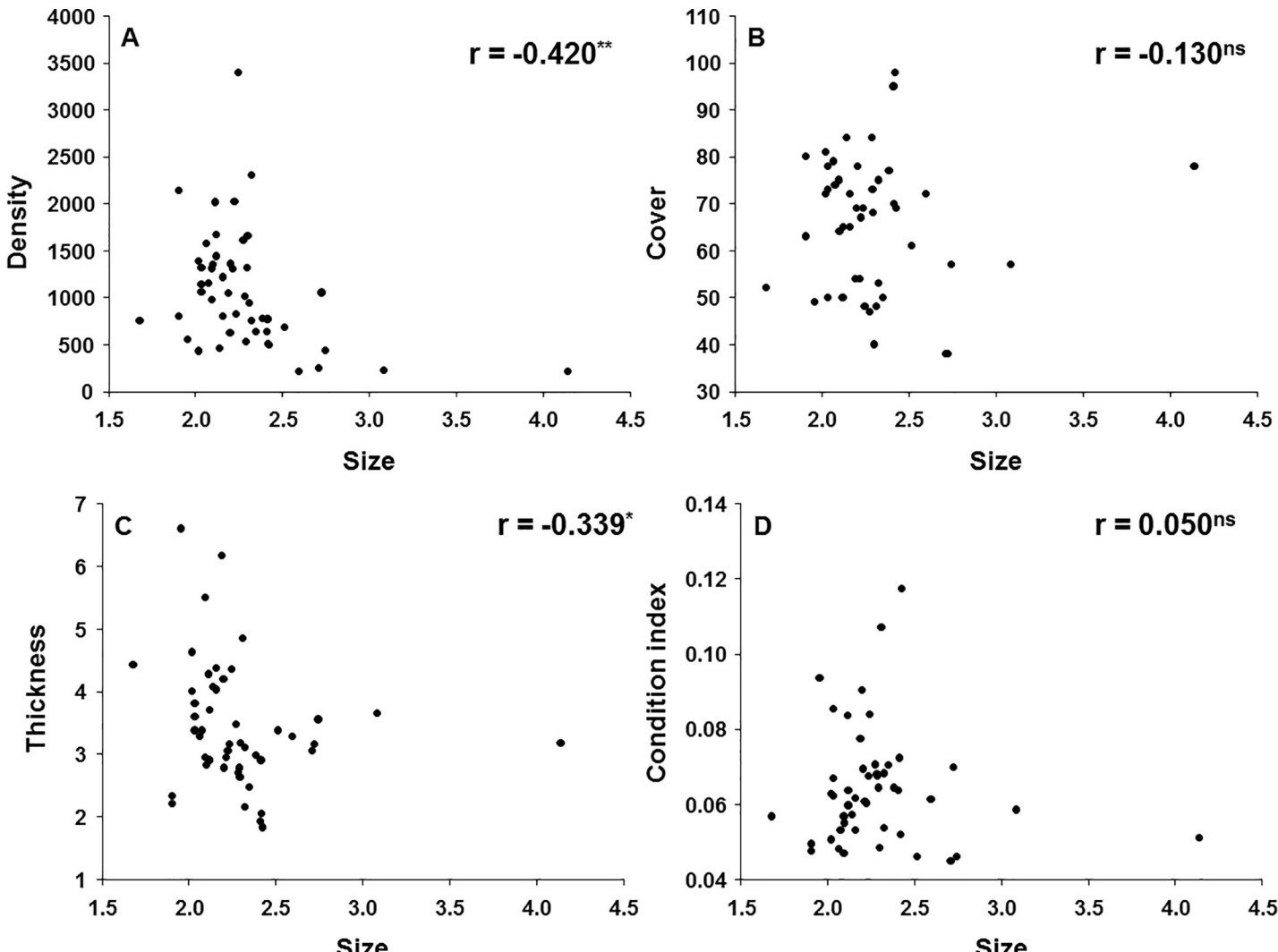

**Fig 5. Values of Spearman's rank correlation coefficient between size and different attributes of mussels.** Density (A), Percentage cover (B), Thickness of clumps (C) and Condition index (D). *: $p < 0.05$; **: $p < 0.01$; ns: Not significant.

length with density and with thickness of clumps. This means that the thickest clumps are composed by a greater number of mussels ($>$ density) but smaller ($<$ shell length). Our results contrast with previous studies done in artificial structures where smaller mussels form thinner beds [47,48]. These differences could be because in our study, sampling was always done on natural rocky substrates that are very different from artificial structures in many characteristics such as slope or material.

Previous studies found that 80% of the individuals of *L. costata*, (a freshwater bivalve) were in the superior half of their size range in the most urbanised areas [42]. These authors considered that recruitment is negatively affected by urbanisation and, consequently, individuals in urban areas are larger. The fishery of another bivalve, the oyster *Ostrea conchaphila*, in British Columbia collapsed in the 1930s and '40s. Its decline and failure to recover was attributed to extremely small recruitment and great juvenile mortality caused by harsh environmental conditions [49]. Previously to its collapse, Stafford [50] alerted that beds of *O. conchaphila* were thin and dominated by larger individuals, a condition interpreted as indicative of fewer recruit

survival. Therefore, our results match with those found for *L costata* and *O. conchaphila* [42,49]. A potential explanation for this could be that in urban areas, recruitment has declined and, as a result, intraspecific competition is smaller, leading to more resources being available for fewer individuals, which can reach larger sizes, as in the urban shores of our study. The significant negative correlation between size and density also supports this. In non-urban areas, density was significantly greater but individuals showed a smaller-size, suggesting more recruitment and enhanced intraspecific competition. The smallest recruitment in urban areas could be the result of either small adult fecundity, small larval survival or great juvenile mortality. An alternative explanation to competition could be differences in predation between urban and non-urban areas. However, these issues should be explored in future manipulative studies.

Philippart et al. [51] indicate that the greatest probability of occurrence of mussel larvae in the Iberian Peninsula was in late winter (end of February) and early fall (end of September). Their results also showed that the presence of mussel larvae is not determined by small-scale (local) variation in environmental conditions, but by environmental conditions that operate at larger (100s and 1000s of km) spatial scales. Therefore, variation in the larval supply does not appear to be responsible for different patterns found in our study between urban and non-urban shores. Considering the fact that in invertebrates the reproductive potential (gonad volume) increases exponentially with size, being larger animals more fecund than small individuals [46,52], and that we found a greater frequency of larger-sized mussels in urban areas, the reproductive capacity should not be the most plausible explanation for our results. However, urbanisation could alter the reproductive process and the importance of fertilisation success should also be considered. At great population density, increased fertilization success could compensate for decreased gamete production. Therefore, small individuals at great population density, as in our non-urban shores, may have similar per capita zygote production as large individuals at small density populations, as in our urban shores [53], providing a reasonable explanation for our results.

Other impacts commonly associated with urban intertidal areas are harvesting and trampling [e.g. 30,31]. Different studies have shown that harvesting causes a reduction in mussel abundance and size [30,44] so harvesting probably is not the main driver responsible for patterns found in our study at least for the largest mussels observed in urban shores. On the other hand, trampling affects mussels directly, by removing all or part of an individual through crushing and dislodgement, or by weakening attachment strength. Thus, it increases the risk of displacement through storms or waves, the susceptibility to predation and the vulnerability of young individuals to the border effect [54, 55]. However, Smith and Murray [26] found a shift towards greater frequencies of smaller individuals as effect of trampling. Therefore, trampling seems not to be a likely driver to explain our results of larger mussel size in urban shores submitted to more human visits (hence more trampling).

Puccinelli et al. [56] assessed if the proximity to urban centres influenced the dietary regime of marine benthic filter feeders. They found that mussels from urbanised sites had fatty acid signatures enriched with a greater proportion of polyunsaturated fatty acids (PUFA), indicative of exposure to large food availability and quality. As human concentration associated to urbanisation promotes nutrient input into the sea, it may also enhance primary production and thus the amount of PUFA in phytoplankton. Puccinelli et al. [56] conclude that urbanisation increases the availability of PUFA for benthic filter feeders and therefore nourishing and suitable food accessibility. This could explain why mussels in urban areas reach a greater-size. Similar effects were found for organic matter supply from fish farms. Mussels close to those farms, below direct organic release, reached greater-sizes than mussels distant from the farms [57,58]. Organic waste from fish farms that disperses in the water column might be a food

resource for mussels, which as filter feeders are essentially generalist consumers of particulate organic matter [59] and exploit organic matter from various sources (autochthonous, terrigenous natural allochthonous or anthropogenic) according to its availability. Thus an organic matter or nutrient increase associated to punctual wastewater seepages in urban areas, dispersed by the strong wave currents, could also explain the greater mussel size observed at urban shores in our study.

Our study used a sampling design including different nested spatial scales from meters to 10s of kilometres and it was done in winter. Previous studies done in spring found also that mussels, in urban areas of the Northern Portuguese coast, reported larger mussel size than at non-urban shores [60]. This means that this pattern seems to be consistent during winter and spring but future studies should explore its consistence along the time.

We considered for our study two variables that measure the abundance of mussels: percentage cover and density. However, we only found significant differences between urban and non-urban conditions for density whereas percentage cover showed significant variability at the scales of shore and site. Moreover, the correlation between both variables was not significant. This means that percentage cover is not a good proxy to estimate the mussel abundance and provides different information (i.e. habitat occupancy). Percentage cover indicates the amount of area occupied by *M. galloprovincialis* whereas density indicates the number of individuals found in a unit area (i.e. 10 cm$^2$). Most of the studies done in rocky intertidal use percentage cover that has the strong advantage of being a non-destructive method, however, our results showed that density seems to be a better descriptor to evaluate the effects of anthropogenic stressors, such as urbanisation. This result should be considered when designing future monitoring programs.

Our results therefore indicate differences in mussel populations, in terms of density and size, between urban and non-urban shores. We speculate that recruitment could be weakened on urban shores and as consequence, intraspecific competition can be also smaller. However, we cannot attribute the observed pattern unambiguously to urbanisation without manipulative studies that will be done in the future. Anyway, as mussel beds are ecosystem-engineers that harbour a great number of individuals and species and provide many ecosystem services to mankind [e.g. 28,29], urbanisation can also have indirect effects on communities associated with mussels and ultimately to human welfare. In this way, evaluating the vulnerability of mussel beds may support assessment and management concerning not only them but also their associated communities and the human welfare.

## Supporting information

**S1 Fig. Distribution of human population along the study area.**
(TIF)

**S1 Table. Full data.**
(XLSX)

## Acknowledgments

We are grateful to the academic editor Gee Chapman, Juan Moreira and two anonymous referees for all the helpful comments and suggestions, which greatly improved this paper.

## Author Contributions

**Conceptualization:** Puri Veiga, Marcos Rubal.

**Data curation:** Catarina Ramos-Oliveira, Leandro Sampaio, Marcos Rubal.

**Formal analysis:** Puri Veiga, Marcos Rubal.

**Funding acquisition:** Puri Veiga.

**Investigation:** Puri Veiga, Catarina Ramos-Oliveira, Leandro Sampaio, Marcos Rubal.

**Methodology:** Puri Veiga, Marcos Rubal.

**Project administration:** Puri Veiga.

**Supervision:** Puri Veiga, Marcos Rubal.

**Validation:** Puri Veiga, Marcos Rubal.

**Visualization:** Puri Veiga.

**Writing – original draft:** Puri Veiga.

**Writing – review & editing:** Puri Veiga, Catarina Ramos-Oliveira, Leandro Sampaio, Marcos Rubal.

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
