## [Decision Letter · Decision Letter 0]

17 Feb 2020

PONE-D-19-34175

The role of urbanisation in shaping the attributes of Mytilus galloprovincialis populations

PLOS ONE

Dear Dr Veiga,

Thank you for submitting your manuscript to PLOS ONE. After careful consideration, we feel that it has merit but does not fully meet PLOS ONE’s publication criteria as it currently stands. Therefore, we invite you to submit a revised version of the manuscript that addresses the points raised during the review process.

Academic Editor

You have three positive reviews of your manuscript, so I am happy to invite you to review the original in the light of these comment s.All comments must be addressed in the revised manuscript or rebutted in an accompanying letter.In the revision please make it clear what revisions have been made to each comment, along with the appropriate line numbers.

In your discussion, make it clear that you have identified differences in the mussel populations between urban and non-urban shores.Even with replication, it is, however, difficult to attribute any differences to urbanization per se.Experiments are needed to address this directly.For example, if urban shores tend to be more sheltered, or have smaller patches of rocky shores (or anything else that one can think up), then differences could be found between mussels on urban or non-urban shores, that have nothing directly to do with urbanization.So in the Discussion, mainly consider the patterns you have found and you can speculate on what may have caused them, but you cannot attribute that unambiguously to urbanization.

Otherwise, the comments are not major, so I expect that you should be able to address them adequately.

We would appreciate receiving your revised manuscript by Apr 02 2020 11:59PM. To enhance the reproducibility of your results, we recommend that if applicable you deposit your laboratory protocols in protocols.io, where a protocol can be assigned its own identifier (DOI) such that it can be cited independently in the future. For instructions see: http://journals.plos.org/plosone/s/submission-guidelines#loc-laboratory-protocols

We look forward to receiving your revised manuscript.

Kind regards,

Maura (Gee) Geraldine Chapman, PhD DSc

Academic Editor

PLOS ONE

Additional Editor Comments (if provided):

Academic Editor

You have three positive reviews of your manuscript, so I am happy to invite you to review the original in the light of these comment s. All comments must be addressed in the revised manuscript or rebutted in an accompanying letter. In the revision please make it clear what revisions have been made to each comment, along with the appropriate line numbers.

In your discussion, make it clear that you have identified differences in the mussel populations between urban and non-urban shores. Even with replication, it is, however, difficult to attribute any differences to urbanization per se. Experiments are needed to address this directly. For example, if urban shores tend to be more sheltered, or have smaller patches of rocky shores (or anything else that one can think up), then differences could be found between mussels on urban or non-urban shores, that have nothing directly to do with urbanization. So in the Discussion, mainly consider the patterns you have found and you can speculate on what may have caused them, but you cannot attribute that unambiguously to urbanization.

Otherwise, the comments are not major, so I expect that you should be able to address them adequately.

Journal Requirements:

2. Your ethics statement must appear in the Methods section of your manuscript. If your ethics statement is written in any section besides the Methods, please move it to the Methods section and delete it from any other section. Please also ensure that your ethics statement is included in your manuscript, as the ethics section of your online submission will not be published alongside your manuscript.

Reviewers' comments:

Reviewer's Responses to Questions

**Comments to the Author**

1. Is the manuscript technically sound, and do the data support the conclusions?

Reviewer #1: Yes

Reviewer #2: Yes

Reviewer #3: Partly

2. Has the statistical analysis been performed appropriately and rigorously? 

Reviewer #1: Yes

Reviewer #2: Yes

Reviewer #3: Yes

3. Have the authors made all data underlying the findings in their manuscript fully available?

Reviewer #1: Yes

Reviewer #2: Yes

Reviewer #3: Yes

4. Is the manuscript presented in an intelligible fashion and written in standard English?

Reviewer #1: Yes

Reviewer #2: No

Reviewer #3: No

5. Review Comments to the Author

Reviewer #1: This manuscript compares populations of the blue mussel, Mytilus galloprovincialis, on “urban” and “non-urban” shores across North Portugal, focusing on cover, abundance and other atttributes of mussels as descriptors. To my understanding, this had not been assessed yet in the Iberian Peninsula in spite of the ecological / economic importance of this bivalve. Sampling consisted in a nested design with several scales of spatial replication, from 10s kms to ms. The statistical analyses are simple, straightforward and (fortunately) things are explained (where they are nested, etc. -not easy to find such information nowadays-). The paper is, in general, easy to follow and might be cited in the short term because of the targeted species and its economic relevance in western Europe.

I have no major concerns apart from some minor issues detailed below, mostly dealing with phrasing and the like; English is not, however, my first (nor second!) language but I recommend the authors to consider my suggestions / recommendations carefully. The paper would benefit from some (just some) rewriting in the Introduction (see comments below) and the main aim as well (as stated in the last paragraph). Some revision/explanation is also needed regarding the using of cover as a proxy of abundance; there is some conflict between the M&M (line 142) and the Discussion (lines 363-370).

ABSTRACT

Check the use of “lower”, “higher” or “lesser number” instead “smaller”, “greater”, “fewer individuals” or “larger” and so on (cfr. lines 26, 30, 31 and other sections of the MS –e.g. line 334).

INTRODUCTION

Line 29: “is declined” (also line 297)

Line 43: “our societies”???

Line 46: “urbanisation is one the toughest, most pervasive and growing menace” --- Please check the phrasing.

Line 54: “…jetties, piers or breakwaters…”

Lines 58-59: “different communities”??? --- Do you mean that the composition of the community is usually different in artificial habitats when compared to natural ones?

Line 59: “algae”

Line 77: “additional adverse effects” ---- So there is/are other effects apart from these “additional” ones.

Line 79: “of their own recruits”

Line 77: “Mussel harvesting” is introduced here; then, “Mussel harvesting” is again treated later in lines 85 and following. Any chance to put all this in sequence?

Lines 88-89: “between density of M. galloprovincialis recruits”

Lines 73-75: “Value of mussels to enhance biodiversity”: This is treated again later (lines 92-96). Again, any chance to put all this together?

Line 98: “may help to its management” or something similar.

Lines 100…: I would state that these mussel attributes are being compared between “non-urban” shores and “urban shores” rather than “to test if urbanisation shapes…”. A similar statement is in the Abstract (lines 19-21). To my understanding, the paper truly provides comparative data among different habitats but I do not see how can be tested if “urbanisation” itself affects mussel polulations as studied here.

Lines 103-104: “Moreover, the relationship between mussel size and other studied attributes will be explored to disentangle potential effects of intraspecific competition.” --- Is this assessed/discussed in the manuscript?

MATERIAL AND METHODS

Line 117: Delete “,” after “W”

Line 118: “they are located” instead of “they occur”

Line 123: Taxonomy of species --- Author is not provided for C. stellatus.

Line 136: “northern”

Line 138: “10s”

Line 142: “galloprovincialis”

Line 142: “abundance” measured as “cover” ---To me, cover is not a proxy of abundance; anyway, this was not justified properly in the text of the M&M, or, alternatively, whether the authors tried to test if it could be indeed such proxy (it is, however, treated in that way in the Discussion, lines 363-370). If the latter, it should be clarified in the M&M and therefore lines 363-370 would make sense. In fact, density (true abundance) was measured after scraping several quadrats per site.

Line 153: You mean ten mussels (mussel = replicate?)

RESULTS

Table 1: “coverture”? --- cover

Figure 5, legend: “Values of Spearman’s rank correlation coefficient”

DISCUSSION

Lines 239-241: Did Gillis et al. find that urbanisation reduced abundance / altered sizes or rather than there was a correlation between this and urbanised areas?

Lines 247-248: “Gillis [43] did not consider nested spatial scales and this could have influenced the results.” --- or just the importance of an appropriate scale of spatial replication.

Line 255 “1758 and” (space)

Line 271: harsher?

Lines 273-275, 315: larger sized? smaller sized?

Line 280: “with a lower size” --- “smaller”, “shortest”???

Line 281: “smaller mussels”

Lines 285-288: Please check the phrasing there.

Line 289: “are larger.”

Line 294-295: “and dominated by larger individuals”, “fewer recruits survive.”

Line 298: “for fewer individuals”

Line 299: “larger sizes”

Line 306: “probability of occurrence of mussel larvae”

Line 311: “larval supply”

Line 319: could compensate?

Line 326 “in mussel abundance and size”

Line 327-328: “for the largest mussels”

Line 328: “On the other hand, trampling affects…”

Line 332: “border effect”

Line 337: “assessed”

Line 341: “As urbanisation promotes nutrient input into the sea,…” --- More likely due to human concentration rather than to urbanisation itself.

Reviewer #2: Title

Suggest to altering to: The role of urbanisation in affecting Mytilus galloprovincialis

The term “attributes” is confusing without any explanation or context, especially in the title

General comments

The manuscript needs to be careful proof-reading, particularly Abstract and Introduction need extra attention. In general the findings have not been successfully discussed in terms of new ideas or the literature. Maybe the findings are not of broad interest because the authors only state a difference with density.

Abstract

I have provided an example of the changes needed to the manuscript, with an example of the Abstract:

See attached

Introduction

Sentences could be simplified, for e.g. the first sentence could read: Marine ecosystems provide food, shoreline protection against storms and floods, water quality maintenance…

Line 46 delete toughest and change menace to threats

Line 78 change unwanted to discarded

Line 79 change diminution to loss

Line 99 start new paragraph with Considering…

Line 103 other studied attributes need to be explained from the beginning as in line 192

Methods and Results

Generally the Methods and Results are written well

Line 109 write NW in full

Line 129 and throughout the manuscript the authors refer to hierarchical sampling but it is in fact a mixed model design with orthogonal and nested factors.

Table 1 change coverture to cover, put the transformation and C-test in the caption

Discussion

The Discussion requires the most work as it this stage. Rather than each paragraph starting with a finding from the literature, it should focus on the findings of the research. The Discussion could also propose a variety of explanations that may explain the observed pattern of linkages with mussels with urbanisation.

Reviewer #3: The manuscript deal with the effect of urbanization on density, percentage cover, depth of clumps, condition index and size frequency distribution of mussels.

The manuscript explored the effect of urbanization, which is really important in the coastal environments and specially in the rocky shores, since it is an important habitat that provide protection for cities. I found the idea original and necessary. I encouraged the authors to review the article and send it again.

The main problem I found on the manuscript is the difficulty in the written. Sometimes it is difficult to understand the idea due to the way it is written. There are also some problems on the concepts and even there are enough data to support some conclusions, others are not supported.

Introduction is clear, except for the objective of the study, that should be rewritten.

The methodology have been conducted properly. The sampling design is rigorous and appropriate.

Abundance and density are used as synonyms in some parts of the manuscript and with different meaning in others. Please, define clearly both terms.

Discussion should be rethought and write. Particularly, the conclusion about competition is not coming from your data. For that, it is necessary to set an experiment testing that hypothesis. Also, conclusions that authors did about nutrients are not part of this manuscript and are confusing leading to several conclusions with the same data. I suggest having fewer conclusions but more clear and based on your data should improve the manuscript.

I attached the manuscript with several comments on it as a complement of this general comment.

6. PLOS authors have the option to publish the peer review history of their article (what does this mean?). If published, this will include your full peer review and any attached files.

Reviewer #1: Yes: Juan Moreira

Reviewer #2: No

Reviewer #3: No

---

## [Author Response · Author response to Decision Letter 0]

26 Mar 2020

Page and line numbers given in the following answers correspond to those of the new re-submitted manuscript (note that these numbers can change after the generation of the PDF version). 

Due to the significant contribution made by the academic editor and referees, we have added the following to the Acknowledgement section:

“We are also grateful to the academic editor Gee Chapman, Juan Moreira and two anonymous referees for all the helpful comments and suggestions, which greatly improved this paper” (Page 17, Line 390-392).

ONE-D-19-34175

The role of urbanisation in shaping the attributes of Mytilus galloprovincialis populations

PLOS ONE

Academic Editor

You have three positive reviews of your manuscript, so I am happy to invite you to review the original in the light of these comments. All comments must be addressed in the revised manuscript or rebutted in an accompanying letter. In the revision please make it clear what revisions have been made to each comment, along with the appropriate line numbers.

1. In your discussion, make it clear that you have identified differences in the mussel populations between urban and non-urban shores. Even with replication, it is, however, difficult to attribute any differences to urbanization per se. Experiments are needed to address this directly. For example, if urban shores tend to be more sheltered, or have smaller patches of rocky shores (or anything else that one can think up), then differences could be found between mussels on urban or non-urban shores, that have nothing directly to do with urbanization. So in the Discussion, mainly consider the patterns you have found and you can speculate on what may have caused them, but you cannot attribute that unambiguously to urbanization.

Otherwise, the comments are not major, so I expect that you should be able to address them adequately.

Thank you for your comments. I have considered your suggestions and those of the three referees. In the discussion, we have added a part to clarify that our patterns can not be unequivocally attributed to urbanisation. (Please see page 17 Lines 376-380).

Comments to the Author

1. Is the manuscript technically sound, and do the data support the conclusions?

Reviewer #1: Yes

Reviewer #2: Yes

Reviewer #3: Partly

2. Has the statistical analysis been performed appropriately and rigorously?

Reviewer #1: Yes

Reviewer #2: Yes

Reviewer #3: Yes

3. Have the authors made all data underlying the findings in their manuscript fully available?

Reviewer #1: Yes

Reviewer #2: Yes

Reviewer #3: Yes

4. Is the manuscript presented in an intelligible fashion and written in standard English?

Reviewer #1: Yes

Reviewer #2: No

Reviewer #3: No

 5. Review Comments to the Author

 

Reviewer #1: This manuscript compares populations of the blue mussel, Mytilus galloprovincialis, on “urban” and “non-urban” shores across North Portugal, focusing on cover, abundance and other atttributes of mussels as descriptors. To my understanding, this had not been assessed yet in the Iberian Peninsula in spite of the ecological / economic importance of this bivalve. Sampling consisted in a nested design with several scales of spatial replication, from 10s kms to ms. The statistical analyses are simple, straightforward and (fortunately) things are explained (where they are nested, etc. -not easy to find such information nowadays-). The paper is, in general, easy to follow and might be cited in the short term because of the targeted species and its economic relevance in western Europe.

2. I have no major concerns apart from some minor issues detailed below, mostly dealing with phrasing and the like; English is not, however, my first (nor second!) language but I recommend the authors to consider my suggestions / recommendations carefully. The paper would benefit from some (just some) rewriting in the Introduction (see comments below) and the main aim as well (as stated in the last paragraph). Some revision/explanation is also needed regarding the using of cover as a proxy of abundance; there is some conflict between the M&M (line 142) and the Discussion (lines 363-370).

We have done the English corrections. The main aim of the paper was also corrected following your suggestion (see page 2 lines 18-21 and page 5 lines 93-98). We compared percentage cover and density of mussels on urban and non-urban shores. In the discussion, we talk about different results obtained for both variables. To avoid the conflict between M&M and discussion, M&M was slightly modified (See page 7 lines 140-149).

ABSTRACT

3. Check the use of “lower”, “higher” or “lesser number” instead “smaller”, “greater”, “fewer individuals” or “larger” and so on (cfr. lines 26, 30, 31 and other sections of the MS –e.g. line 334).

We agree with referee and this was corrected along de manuscript.

INTRODUCTION

4. Line 29: “is declined” (also line 297)

We have changed to “has decline”.

5. Line 43: “our societies”???

We agree. The change was done.

6. Line 46: “urbanisation is one the toughest, most pervasive and growing menace” --- Please check the phrasing.

This was modified to “urbanisation is one of the most pervasive and growing threats”.

7. Line 54: “…jetties, piers or breakwaters…”

The change was done.

8. Lines 58-59: “different communities”??? --- Do you mean that the composition of the community is usually different in artificial habitats when compared to natural ones?

Yes, we do. “different communities” was change to “different community composition”.

9. Line 59: “algae”

Done.

10. Line 77: “additional adverse effects” ---- So there is/are other effects apart from these “additional” ones.

Additional was deleted.

11. Line 79: “of their own recruits”

Done.

12. Line 77: “Mussel harvesting” is introduced here; then, “Mussel harvesting” is again treated later in lines 85 and following. Any chance to put all this in sequence?

We understand your comment. In page 4, lines 78-81 content was modified because in this part we talk about disappearance of intertidal beds of M. edulis in 1990 in the Dutch Wadden Sea as consequence of recruitment failure, intensive fishery and natural mortality, not only from harvesting.

13. Lines 88-89: “between density of M. galloprovincialis recruits”

Done.

14. Lines 73-75: “Value of mussels to enhance biodiversity”: This is treated again later (lines 92-96). Again, any chance to put all this together?

In lines 73-75, we treat the role of mussels increasing the biodiversity. Between lines 92 and 96, we talk about different ecosystem services provided by mussel beds in general. 

15. Line 98: “may help to its management” or something similar.

Done.

16. Lines 100…: I would state that these mussel attributes are being compared between “non-urban” shores and “urban shores” rather than “to test if urbanisation shapes…”. A similar statement is in the Abstract (lines 19-21). To my understanding, the paper truly provides comparative data among different habitats but I do not see how can be tested if “urbanisation” itself affects mussel polulations as studied here.

Done.

17. Lines 103-104: “Moreover, the relationship between mussel size and other studied attributes will be explored to disentangle potential effects of intraspecific competition.” --- Is this assessed/discussed in the manuscript?

Yes. You can see in page 14 and lines 294-299 where we elucidate about results of Spearman's rank correlations and the “plausible” role that intraspecific competition could play explaining patterns found in our study. 

MATERIAL AND METHODS

18. Line 117: Delete “,” after “W”

Done.

19. Line 118: “they are located” instead of “they occur”

Done.

20. Line 123: Taxonomy of species --- Author is not provided for C. stellatus.

Done.

21. Line 136: “northern”

Done.

22. Line 138: “10s”

Done.

23. Line 142: “galloprovincialis”

Done.

24. Line 142: “abundance” measured as “cover” ---To me, cover is not a proxy of abundance; anyway, this was not justified properly in the text of the M&M, or, alternatively, whether the authors tried to test if it could be indeed such proxy (it is, however, treated in that way in the Discussion, lines 363-370). If the latter, it should be clarified in the M&M and therefore lines 363-370 would make sense. In fact, density (true abundance) was measured after scraping several quadrats per site.

See response to comment 2.

25. Line 153: You mean ten mussels (mussel = replicate?)

Yes. Replicate was changed to mussels.

RESULTS

26. Table 1: “coverture”? --- cover

Done.

27. Figure 5, legend: “Values of Spearman’s rank correlation coefficient”

Done.

DISCUSSION

28. Lines 239-241: Did Gillis et al. find that urbanisation reduced abundance / altered sizes or rather than there was a correlation between this and urbanised areas?

This part was modified, please see page 11 and lines 237-239.

29. Lines 247-248: “Gillis [43] did not consider nested spatial scales and this could have influenced the results.” --- or just the importance of an appropriate scale of spatial replication.

This part was modified, please see page 12 lines 244-246.

30. Line 255 “1758 and” (space)

Done.

31. Line 271: harsher?

Done.

32. Lines 273-275, 315: larger sized? smaller sized?

Done.

33. Line 280: “with a lower size” --- “smaller”, “shortest”???

Done.

34. Line 281: “smaller mussels”

Done.

35. Lines 285-288: Please check the phrasing there.

This part was modified. Please check page 13 lines 284-287.

36. Line 289: “are larger.”

Done.

37. Line 294-295: “and dominated by larger individuals”, “fewer recruits survive.”

Done.

38. Line 298: “for fewer individuals”

Done.

39. Line 299: “larger sizes”

Done.

40. Line 306: “probability of occurrence of mussel larvae”

Done.

41. Line 311: “larval supply”

Done.

42. Line 319: could compensate?

Done.

43. Line 326 “in mussel abundance and size”

Done.

44. Line 327-328: “for the largest mussels”

Done.

45. Line 328: “On the other hand, trampling affects…”

Done.

46. Line 332: “border effect”

Done.

47. Line 337: “assessed”

Done.

48. Line 341: “As urbanisation promotes nutrient input into the sea,…” --- More likely due to human concentration rather than to urbanisation itself.

This was clarified.

Reviewer #2: Title

49. Suggest to altering to: The role of urbanisation in affecting Mytilus galloprovincialis. The term “attributes” is confusing without any explanation or context, especially in the title.

We agree with the referee and the title was changed.

General comments

50. The manuscript needs to be careful proof-reading; particularly Abstract and Introduction need extra attention. In general the findings have not been successfully discussed in terms of new ideas or the literature. Maybe the findings are not of broad interest because the authors only state a difference with density.

The manuscript was carefully checked. We have done all the English corrections proposed by referees and a fluent English speaker has revised the final version. We disagree about the findings have not been successfully discussed in terms of new ideas or the literature. Our study is novelty because in the marine realm M. galloprovincialis has never been compared between urban and non-urban shores. In view that urbanisation is one of the most critical and widespread threats to coastal marine ecosystems we think that our study is pertinent. About literature, we did an exhaustive bibliographic search to discuss our results. If the referee knows of some references that have not been considered please tell us which were missed? 

Regarding the issue that “findings are not of broad interest because authors only state a difference with density”. We found also differences in terms of size (mussels showed a greater frequency of larger individuals on urban shores). Moreover, the lack of differences can be as relevant as significant differences for instance in management issues. 

Abstract

51. I have provided an example of the changes needed to the manuscript, with an example of the Abstract:

See attached

Done.

Introduction

52. Sentences could be simplified, for e.g. the first sentence could read: Marine ecosystems provide food, shoreline protection against storms and floods, water quality maintenance…

Done. See response to comment 50.

53. Line 46 delete toughest and change menace to threats

Done.

54. Line 78 change unwanted to discarded

Done.

55. Line 79 change diminution to loss

Done.

56. Line 99 start new paragraph with Considering…

Done.

57. Line 103 other studied attributes need to be explained from the beginning as in line 192

This was clarified (see page 5 line 97).

Methods and Results

Generally the Methods and Results are written well.

58. Line 109 write NW in full

Done.

59. Line 129 and throughout the manuscript the authors refer to hierarchical sampling but it is in fact a mixed model design with orthogonal and nested factors.

This mistake was corrected along the manuscript.

60. Table 1 change coverture to cover, put the transformation and C-test in the caption

In table 1, coverture was changed to cover but we prefer maintain transformation and C-test in the table instead in the caption.

Discussion

61. The Discussion requires the most work as it this stage. Rather than each paragraph starting with a finding from the literature, it should focus on the findings of the research. The Discussion could also propose a variety of explanations that may explain the observed pattern of linkages with mussels with urbanisation.

Discussion was also modified following recommendation of referees and the editor. We started the discussion with our results in the first part of the discussion “The most notorious result of our study was that mussels at urban shores showed a smaller density and a greater frequency of larger individuals”. Please see page 11 lines 233-235. Then we elucidate potential explanations for our observed patterns such as differences on recruitment, intraspecific competition, predation or outfalls between urban and non-urban shores.

Reviewer #3: The manuscript deal with the effect of urbanization on density, percentage cover, depth of clumps, condition index and size frequency distribution of mussels. The manuscript explored the effect of urbanization, which is really important in the coastal environments and specially in the rocky shores, since it is an important habitat that provide protection for cities. I found the idea original and necessary. I encouraged the authors to review the article and send it again.

62. The main problem I found on the manuscript is the difficulty in the written. Sometimes it is difficult to understand the idea due to the way it is written. There are also some problems on the concepts and even there are enough data to support some conclusions, others are not supported.

The manuscript was carefully checked. We have done all the English corrections proposed by referees and a fluent English speaker has revised the final version.

63. Introduction is clear, except for the objective of the study, that should be rewritten. The methodology have been conducted properly. The sampling design is rigorous and appropriate.

The objective of the study was modified according to referee 1. See page 2 lines 18-21 and page 5 lines 93-98 

64. Abundance and density are used as synonyms in some parts of the manuscript and with different meaning in others. Please, define clearly both terms.

This was corrected and clarified. Please check response to comment 1.

65. Discussion should be rethought and write. Particularly, the conclusion about competition is not coming from your data. For that, it is necessary to set an experiment testing that hypothesis. Also, conclusions that authors did about nutrients are not part of this manuscript and are confusing leading to several conclusions with the same data. I suggest having fewer conclusions but more clear and based on your data should improve the manuscript.

This part was modified to be little assertive. However, we found a significant negative correlation between density of mussels and size. If density is lower in urban shores and mussels are of larger size, as our results showed, then intraspecific competition seem to be smaller in urban shores. About nutrients, it is only a plausible explanation for the greater frequency of larger size mussels. It is well known that urban areas usually present outfalls that can provide organic matter, nutrients, among others. Nutrients can increase the primary production and the availability of polyunsaturated fatty acids for filter feeders, as mussels, that could incorporate them as a food item.

66. I attached the manuscript with several comments on it as a complement of this general comment.

We have done and considered all your corrections and comments.

About your comment “As far as I understand from the methodology, you measured abundance and note density. Authors used density or abundance with the same meaning. Please, check the terms and used only one”. 

Please see response to comment 2.

---

## [Editor Report · Decision Letter 1]

22 Apr 2020

The role of urbanisation in affecting Mytilusgalloprovincialis

PONE-D-19-34175R1

Dear Dr. Veiga,

We are pleased to inform you that your manuscript has been judged scientifically suitable for publication and will be formally accepted for publication once it complies with all outstanding technical requirements.

With kind regards,

Maura (Gee) Geraldine Chapman, PhD DSc

Academic Editor

PLOS ONE
---

## [Editor Report · Acceptance letter]

29 Apr 2020

PONE-D-19-34175R1 

The role of urbanisation in affecting Mytilus galloprovincialis 

Dear Dr. Veiga:

I am pleased to inform you that your manuscript has been deemed suitable for publication in PLOS ONE. Congratulations! Your manuscript is now with our production department. 

With kind regards,

on behalf of

Professor Maura (Gee) Geraldine Chapman 

Academic Editor

PLOS ONE